# Current Knowledge about Headaches Attributed to Ischemic Stroke: Changes from Structure to Function

**DOI:** 10.3390/brainsci13071117

**Published:** 2023-07-23

**Authors:** Xinxin Xie, Yi Zhang, Qianqian Kong, Hao Huang, Zhiyuan Yu, Xiang Luo, Wensheng Qu

**Affiliations:** 1Neurological Department, Tongji Hospital, Tongji Medical College, Huazhong University of Science and Technology, Wuhan 430030, China; xxx980309@163.com (X.X.); zhangi0315@163.com (Y.Z.); kqq0203@163.com (Q.K.); huanghtj@gmail.com (H.H.); zhiyuan_yu@126.com (Z.Y.); flydottjh@163.com (X.L.); 2Hubei Key Laboratory of Neural Injury and Functional Reconstruction, Huazhong University of Science and Technology, Wuhan 430030, China

**Keywords:** Ischemic stroke, headache, structure, function, clinical features, pathophysiology, imaging technique, signal transducers

## Abstract

Headaches are common after ischemic stroke (IS). Unlike primary headaches, headaches attributed to IS have specific clinical features. This review describes the epidemiology, clinical characteristics, risk factors, and influence of IS headaches. Previous reports were summarized to show the correlations between headaches and structural lesions in the cerebral cortex, subcortical white matter, deep gray matter nuclei, brainstem, and cerebellum. However, the substantial heterogeneity of IS, subjective evaluations of headaches, and inadequate cohort studies make it difficult to explore the pathophysiology of headaches attributed to IS. In our recommendation, favorable imaging techniques, such as magnetic resonance imaging and positron emission tomography, may provide new insights into mechanical studies of IS headaches from structure to function. It may also be helpful to extend the research field by targeting several shared signal transducers between headaches and IS. These markers might be neuropeptides, vasoactive substances, ion channels, or electrophysiologic changes.

## 1. Introduction

Headaches are a common complication of ischemic stroke (IS) that is often overlooked. It can also cause disability through severe attacks or when it has been combined with other disabling impairments [1,2]. According to the 2018 International Classification of Headaches Disorders, 3rd Edition (ICHD-3) [3], headaches attributed to IS/cerebral infarction are classified as a type of secondary headaches (code 6.1.1). It develops simultaneously, or in a close temporal relationship, with signs or other evidence of IS accompanied by neuroimaging confirmation of ischemic infarction. Depending on the course of the headaches, they are divided into acute (new and usually acute onset, code 6.1.1.1) and persistent (lasts more than 3 months after the stroke has stabilized, code 6.1.1.2) subtypes. Acute headaches are usually self-recoverable; however, persistent headaches can become a prominent feature of IS, resulting in a painful condition and even causing disability in daily life [4].

To be attributed to IS, the headaches develop in very close temporal relation to other symptoms and/or clinical signs of IS and usually significantly improve in parallel with stabilization or improvement in these IS features [3]. To date, the understanding of IS headaches has been limited or even controversial. The characteristics of the headaches are variable. They can be ipsilateral to the stroke or bilateral; they can present as a migraine or tension-type but can also present with an isolated sudden (even thunderclap) type in some rare conditions [5]. They are more frequent in basilar-territory than in carotid-territory strokes. Although rarely associated with lacunar infarcts, headaches are also popular in these patients [6].

The heterogeneity of IS, the variable characteristics of headaches in nature, and the reports from different cohorts have made it difficult to identify this type of headache. Moreover, current studies are still insufficient to elucidate the clinical features and pathogenesis of headaches specifically attributed to IS. Therefore, we organized this review to extend our knowledge about headaches attributed to IS and propose future research strategies using favorable techniques.

## 2. Methods

With the assistance of an academic librarian, one of our authors (X. Xie) searched five electronic databases: Web of Science, PubMed, ScienceDirect, OVID Technologies, and Wiley Online Library. These searches were supplemented by tracking the references of interests from included original reports and relevant reviews. Searches were run on 16 May 2023.

We sought to identify all the studies reporting the relationship between headaches and IS. The keywords IS, headache, and prevalence/epidemiology/risk factor/characteristics/symptom/influence were used to search the features of IS headache; since the specific reports for pathophysiology are few, IS/headaches and cerebral cortex/subcortical white matter/thalamus/hypothalamus/brainstem/cerebellum were used as searching keywords to discuss the mechanisms of site-related headache; structural and functional imaging methods were collected from the above reports; at last, reports of headaches-associated neuropeptides, vasoactive substances, ion channels, and electrophysiologic changes were summarized. The literature has never targeted IS, or headaches were excluded.

## 3. Features of IS Headache

### 3.1. Epidemiology

The exact prevalence of headaches attributed to IS is still unclear due to discrepancies in research populations, locations, and sample sizes. A meta-analysis used 20 prospective studies (from 2009 to 2015) to estimate the prevalence of headaches after IS, which resulted in a pooled prevalence of 0.14 (95% confidence interval [CI], 0.07–0.23). Headaches frequency range from 6% to 44% of the IS population [5]. Regional disparity also existed, suggested by a higher prevalence in Europe (0.22, 95% CI, 0.14–0.30) and North America (0.15, 95% CI, 0.05–0.26) than in the Middle East and Asia (0.08, 95% CI, 0.01–0.18) [5,7]. Summarizing reports from 1978 to 2020, the frequency of headaches at the onset of acute IS ranged from 8% to 34% [8,9,10]. Female patients were more likely to develop headaches at stroke onset (odds ratios [OR], 1.3; 95% CI, 1.1–1.6) than male patients. Regarding age, patients aged 40 years had a 4.2-fold increased odds (95% CI, 2.6–6.8) of experiencing headaches at stroke onset compared with patients aged 80 years and older [8]. A few reports included observations of persistent IS headaches. The prevalence of any persistent poststroke headaches ranged from 7–23%, with follow-up times ranging from 3 months to 3 years [4]. In our unpublished observational study, 12.0% of cases reported persistent headaches in the third month after IS. However, this may be grossly underestimated since some patients are unconscious, aphasic, or unable to express their headache symptoms after IS.

### 3.2. Clinical Characteristics

In a study of 550 patients with first-ever IS, approximately 14.9% experienced a headache at stroke onset, and 56% experienced a new type of headache simultaneously with stroke onset [11]. Most patients reported nonspecific features of the headache, often described as tension-like, migraine-like, throbbing, or persistent nonpulsating in nature. Most headaches were tension-like (50–80%) [5,7] and usually mild [12], although moderate to severe pain was also reported [13]. Accompanying symptoms were nausea/vomiting (23–40%) and phonophobia/photophobia (24–36.7%) [7,14]. Dizziness (55.6%) was also a frequent symptom accompanying persistent IS headaches in our unpublished observation. The pain was located in the frontal (67%), temporal (63%), occipital (53%), and neck areas (31%) [8]; 60–65% of headaches were unilateral and were usually ipsilateral [8,14]. Altered features compared with those of previous headaches are necessary for attributing headaches to IS, which might include increased duration (70%); change in location (26.7%); change in the side of the headaches (20%); increased intensity (36.7%); and new concomitant symptoms, such as nausea, vomiting, photophobia, and phonophobia (36.7%) [11].

### 3.3. Risk Factors

In a recent meta-analysis, the female sex showed a modestly higher risk of IS headaches (pooled OR, 1.25; 95% CI, 1.07–1.46); however, an age-dependent difference was not reported [5]. Patients with a previous migraine condition (OR, 6.7; 95% CI, 1.4–31.2) [15], lack of sleep (OR, 1.85; 95% CI, 1.01–3.38), and low physical activity (OR, 2.4; 95% CI, 1.34–4.31) [11] were more likely to suffer from IS headaches. Headaches initiated at stroke onset were associated with cardioembolism (OR, 2.4; 95% CI, 1.4–4.1), posterior circulation stroke (OR, 2.0; 95% CI, 1.2–3.5), large infarcts (size > 15 mm, 2.08-fold; 95% CI, 1.1.−2.7), infarcts of the cerebellum (OR, 2.3; 95% CI, 1.1–4.8), good neurological status (OR, 2.5; 95% CI, 1.2–4.9), and a low frequency of large-artery atherosclerosis (OR 0.4; 95% CI. 0.2–0.8) [11]. A meta-analysis reported a higher prevalence (pooled OR, 1.92; 95% CI, 1.4–2.64) of IS headaches in posterior circulation strokes, which is a widely unknown risk factor. For treatments, endovascular thrombectomy potentiated de novo headaches after IS. In a study of 170 patients after thrombectomy who completed an ad hoc questionnaire, 26.5% (95% CI, 18.8–35.5%) had headaches related to the thrombectomy [16]. A prospective observational cohort study was performed in our stroke center: headaches were observed in 25.0% (138/234) of patients after ischemia, while intravenous thrombolysis (OR, 2.51; 95% CI, 1.313–4.782) increased the presence of persistent headaches attributed to IS.

### 3.4. Influence

Although IS headaches are usually of mild to moderate severity, some patients may experience disabling pain. In a 3-year follow-up study, 62.5% of patients reported headaches of moderate to severe intensity, and medication overuse occurred in 6.25% of cases [7]. However, the early presence of headaches is not as worrisome because it may predict a better prognosis of stroke. IS patients with headaches reportedly had less deterioration upon hospitalization, less neurological dysfunction, fewer complications, and a better prognosis [17]. In contrast, other studies have shown no influence or even worsening relationships between these headaches and early neurological deterioration [18]. It is speculated that headaches in young patients with mild symptoms might be related to collateral circulation reconstruction, indicating a good prognosis, while severe headaches indicate tissue compression and structural changes, which indicates poor prognosis [19]. Chronic headaches were also reported as an independent predictor for better cognitive scores post-stroke [20].

## 4. Structural and Functional Changes During IS Headache

Headaches occur when the abnormality of endogenous pain modulation has been upset by external risk factors. This pain modulation system is organized by trigeminovascular, brainstem, thalamic, and hypothalamic neurons; modulation of cortical brain regions; and activation of descending pain circuits [21]. Studies have tried to identify the different clinical features of IS headaches in specific infarct sites. It was reported that a pulsating headache quality was associated with widespread cortical and subcortical lesion clusters in the posterior and frontotemporal regions, and phonophobia was associated with cerebellar lesions [22]. However, the substantial heterogeneity of IS makes it difficult to identify the correlations among these factors. Reviewing headache-related structural and functional changes after focal lesions may allow us to routinely explore the pathophysiology of IS headaches.

### 4.1. Cerebral Cortex, Especially the Insular Cortex

Headaches are common in patients with cortical stroke (56.5%) [23]. The prevalence of headaches are 29% in those with cortical infarcts vs. 12% in those with small deep infarcts [24]. Direct ischemic insults on meningeal pain-sensitive structures are believed to mediate the occurrence of headaches; however, the understanding of this mechanism is still inadequate. A disease commonly insulting cerebral cortical areas, known as mitochondrial encephalomyopathy, lactic acidosis, and stroke-like episodes (MELAS) syndrome, may serve as a model for researching cortical IS headaches. Recurrent headaches occur in 54–91% of individuals with MELAS. Migrainous headaches in the form of recurrent attacks of severe pulsatile headaches with frequent vomiting are typical and are usually more severe during stroke-like episodes [25,26]. Some triggering metabolic changes in the cerebral cortex may lower the threshold of cortical excitability and cause neuronal hyperexcitability, activating the trigeminovascular fibers that innervate cortical blood vessels and causing headaches [26].

In addition, headaches also occurred in patients with infarcts involving insular and somatosensory cortical brain regions; headache intensity was associated with patients with strokes involving the posterior insula and operculum; and cranial autonomic symptoms accompanying headaches were found in patients with strokes affecting the parietal lobe, somatosensory cortex, and middle temporal cortex [5,22].

As the so-called ‘pain matrix’, the insular cortex is involved in the processing of painful somatosensory inputs [27]. Its anterior region is involved in the emotional and attentional processing of pain [28], whereas the posterior region is a key area for encoding pain intensity [29]. Functional dissociation of latency coding and intensity coding of painful stimuli in the anterior and posterior regions of the insula has been proposed in chronic pain [30]. The inputs from the thalamus and its functional connection to the premotor, sensorimotor, supplementary motor, and cingulate cortex support the integrative role of the insula in sensory processing [31].

In cases with a first-ever acute stroke restricted to the insula, headaches appeared as one atypical presentation in addition to motor and somatosensory deficits, dysarthria, aphasia, and a vestibular-like syndrome. However, details of the headaches were not provided [32]. In primary headaches, a reduced volume of insular gray matter [33] and disrupted functional connectivity networks [34] have been demonstrated. This may similarly occur after local ischemic lesions and lead to headaches. Reportedly, patients with posterior insular strokes and strokes of the opercular-insular complex tend to have higher pain ratings [22]; headaches in patients with acute IS are often associated with infarcts in the anterior and posterior insular cortex [14].

### 4.2. Subcortical Cerebral White Matter

The cortical areas and subcortical structures are connected by ascending and descending tracts of white matter (WM). The clinical correlations between subcortical WM lesions and headaches were previously presented [35]. Migraine is a widely documented independent risk factor for subclinical focal deep WM lesions; WM lesions are commonly observed in migraine patients with aura [36]. Since there is increased cerebral vulnerability to ischemia in migraineurs and blood-brain barrier disruption occurs during migraine attacks, both ischemic and inflammatory mechanisms have been proposed, and spreading depolarization and other electrovascular events may also be involved [36].

In one Japanese study of non-stroke subjects [37], headache attacks between patients with silent stroke and those with deep WM lesions were compared. The results suggested that deep WM lesions, rather than silent brain infarcts, appear to be more likely to produce headaches of nonspecific cause. In a general population-based study [35], individuals with tension-type headaches were more likely to have extensive WM hyperintensities than headache-free subjects (Scheltens scale: OR, 2.46; 95% CI, 1.44–4.20); those with new-onset headaches were more likely to have extensive WM hyperintensities than those who were stable and headache-free (Scheltens scale: OR, 2.24; 95% CI, 1.13–4.44).

However, current reports are unable to show the correlation between subcortical IS and headaches since WM lesions are usually attributed to chronic small vessel diseases. Additionally, although subcortical infarctions present incidentally in certain conditions, their contribution to secondary headaches is still controversial. Therefore, observational studies are strongly suggested in the future.

### 4.3. Deep Gray Matter Nuclei

The thalamus is a nuclear complex in the central deep brain structure of the diencephalon between the midbrain and cortex. Its principal function is to relay and modulate sensory and motor information, including pain regulation [38]. It also plays an important role in communication through the transthalamic transfer of information between cortical areas [39]. Lesions within the ventrocaudal regions of the thalamus carry the highest risk of developing pain. Thalamic pain is a severe and treatment-resistant type of central pain that may develop after thalamic stroke. Although headaches appear as an atypical symptom after thalamic IS, no cohort study for this type of headache has been performed. Stabbing headaches had been described in a case report of a patient secondary to a thalamic hemorrhage [40]. In migraine studies, structural and functional findings suggest abnormal connectivity between the thalamus and various cortical regions, indicating altered pain processing, which plays a critical role in migraine-related allodynia and photophobia [41].

The hypothalamus is a fairly small brain region located around the third ventricle of the brain. It regulates body temperature, sleep, food/water intake, the autonomic nervous system, and cyclical phenomena, such as circadian rhythms and periodic body rhythms [42]. Abnormalities in the hypothalamus underlie the typical premonitory symptoms of migraine, such as yawning, fatigue, appetite, and nausea [43]. Another headache type, cluster headache, clinically involves hypothalamic abnormalities and has been identified to play a crucial role in attack generation [42].

However, unlike autoimmune diseases, IS occurring in the hypothalamus is not as common in the clinic. However, this does not negate the role of the hypothalamus in headaches attributed to stroke, considering the hypothalamic involvement in the neuronal network for headaches. The spontaneous oscillations of complex networks, which involve the hypothalamus, brainstem, and dopaminergic networks, lead to changes in susceptibility thresholds that start but also ultimately terminate headache attacks [42]. Neuronal projections from the basal ganglia and hypothalamus reach the posterior and lateral posterior thalamic nuclei and then the forebrain. This was speculated to play a role in migraine attacks triggered by disrupted sleep, skipping meals, and emotional reactions [43]. Of the deep gray matter nuclei, the thalamus and hypothalamus are structures crucially related to pain. A report showed that ischemic lesions in the lateral thalamus were related to nausea, a common symptom in both stroke and headache, according to magnetic resonance imaging (MRI) voxel-based symptom lesion mapping [22]. However, the current reports are not nearly sufficient to speculate on the influence of deep gray matter nuclei stroke on resulting headaches.

### 4.4. Brainstem and the Cerebellum

Many studies have revealed the role of brainstem structure and function in headaches [44]. Modulation of specific brainstem nuclei alters sensory processing related to these symptoms, including headache, cranial autonomic responses, and homeostatic mechanisms [44]. These symptoms involve some critical nuclei, such as the nucleus pulposus, trigeminal nucleus caudalis, trigeminocervical complex, supraoptic salivary nucleus, locus coeruleus, nucleus raphe magnus, parabrachial nucleus, dorsal raphe nucleus, and periaqueductal gray. These structures comprise an “endogenous pain modulation pathway” with complicated neural networks and neurotransmitters [21]. In a cohort of 387 patients with acute lacunar infarction, headaches were relatively common in the brainstem (mesencephalon and pons) [6]. However, headaches attributed to brainstem IS have not been commonly observed in clinical practice. As reported in a cohort of 2196 patients who experienced IS or transient ischemic attack, headaches at stroke onset were positively associated with cerebellar stroke but not with brainstem locations [8].

The cerebellum is known to be associated with migraine initiation, symptom generation, and headaches [45]. Specific cerebellar dysfunction via genetically driven excitatory/inhibitory imbalances, oligemia, and/or increased risk of WM lesions has been proposed as a critical contributor to migraine pathogenesis [46]. Headaches are the most common initial symptom after cerebellar IS. For acute headaches, symptoms occur when the pain-sensitive structure has been stimulated by tissue edema and compression due to the increased pressure in the posterior cranial fossa. In a 200-patient retrospective observation study, 11% of cerebellar IS patients experienced headaches at stroke onset; more headaches occurred if the IS was in the left hemisphere (60.9%), cortical/juxtacortical area (63.6%), or ischemia of the posterior inferior cerebellar artery perfusion territory (54.5%), and this type of headaches was associated with low morbidity unless there was a conscious disorder [47]. Cerebellar strokes frequently cause nausea via lesions of central vestibular projections [22].

## 5. Research Recommendations for Headaches Attributed to IS

As documented above, the literature clarifying the relationship between IS and subsequent headaches is still limited. Neuroimaging has provided contributions because it is a noninvasive method for bridging our understanding between structural and functional changes. Imaging techniques, including MRI and positron emission tomography (PET), are preferable for detecting the correlations between IS and headaches. They have been recommended to perform detailed clinical history recording and relevant clinical examinations. Conspicuously, the progression of IS shares many signal transducers that are also important for the propagation of headaches. These hotspots may help us to understand the pathophysiology of IS headaches.

### 5.1. Structural Imaging

Structural MR usually includes T2 sequences, fluid attenuation reverse recovery, diffusion-weighted imaging, and high-resolution T1 sequences. By undergoing brain MRI examinations, lesions as regions of interest can be drawn on 3D images and transformed into standard stereotaxic space. Then, the associations between lesion location and the subsequent headaches characteristics can be measured by voxel-based symptom lesion mapping [14,22].

Tractography can also be performed using a different type of MRI called diffusion tensor imaging (DTI). This approach examines the direction of water flow in a voxel and is particularly well suited to mapping the integrity and direction of myelin sheaths. In practice, DTI data are analyzed with tract-based spatial statistics and automated tractography to detect microstructural changes in WM [48]. Hence, this is a convenient approach for mapping the structural correlations between subcortical infarction and headaches.

### 5.2. Functional Imaging

Structures are usually depicted with functional imaging in exploring headaches pathophysiology. Among these methods, functional MRI (fMRI) has played a key role. For instance, fMRI has detected spreading cortical depolarization by tracing changes in oxygenated blood flow [49]. fMRI maps the networks involved in headache-related brain activity, which could potentially identify the IS-induced disconnection of cerebral networks. Usually, resting-state fMRI has a great advantage in this research [50]. Another MR method, MR spectroscopy, allows the investigation of neuronal and glial integrity and metabolism in vivo [51]. For example, abnormal energy metabolism and excitatory–inhibitory balance were confirmed in migraineurs by measuring the ratios of N-acetylaspartate (NAA)/creatine (Cr), choline (Cho)/Cr, and myoinositol (MI)/NAA [52]. This approach should be feasible for detecting the neurochemical metabolite changes in infarction that are attributed to headaches.

Conspicuously, PET helped to identify the dorsal parts of the brainstem as “the migraine generator” [53]. Using H_2_^15^O-labeled PET, the activation of brain regions (the subcortical area and the brainstem) was detected during spontaneous headaches [54]. Using ligand PET, e.g., a radioligand antagonist of 5-HT(1A) receptors ((18)F-MPPF), binding potential can reflect an increase in receptor density or a decrease in endogenous serotonin, which could explain the altered cortical excitability in headaches [55]. Regional cerebral blood flow (rCBF) can be used as a marker of neuronal activity. Significant changes in rCBF related to IS could be highly suggestive of the impact of structural lesions on the pathophysiology of the subsequent headaches [56]. These PET methods may demonstrate that headaches could be generated by IS lesions.

### 5.3. Recording the Signal Transducers

A wide range of neuropeptides and other vasoactive substances are important for pain transmission. These include calcitonin gene-related peptide (CGRP), substance P (SP), pituitary-adenylate-cyclase-activating polypeptide (PACAP), neurokinin A (NKA), neurokinin A, and others [57]. CGRP is a vasoactive peptide released from sensory nerves and is possibly involved in attenuating cerebral ischemia by promoting cell proliferation, reducing neuroinflammation, and reducing vasodilatory action or indirectly through the restoration of blood–brain barrier dysfunction [58]. Similarly, PACAP has been proven to be neuroprotective in IS by protecting against numerous toxic insults, reducing hippocampal neuronal death, and inducing tolerance of cortical neurons subjected to ischemia [59]. It is of interest whether these elevated responding neuropeptides after ischemia could promote the appearance of secondary headaches. However, plasma tests are highly variable and have failed to associate plasma CGRP levels with collateral flow or the prediction of clinical outcomes [60]. Cerebrospinal fluid (CSF) specimens in human or animal tissue measurements may provide stronger evidence.

Ion channels and mechanotransducers are important for membrane depolarization, action potential generation, propagation, and neurotransmitter release. Several types of TRP channels and K^+^ channels play an important role in neuronal excitation [21]. Voltage-gated calcium channels (VGCCs) play a unique role in migraine. VGCC dysfunction is also involved in the development of acute IS. The increase in α2δ-1 subsites in serum and CSF specimens may be used as a potential marker for VGCC dysfunction, illness severity, and neuroinflammation in IS patients [61]. Adenosine triphosphate (ATP) and low pH can sensitize chemosensitive neurons following injuries [21]. ATP released after cell disruption activates glial cells and enhances neuroinflammation. Acid-sensing ion channels (ASICs) responding to decreased extracellular pH are associated with hypoxia, injury, or inflammation. Decreases in meningeal pH can evoke ASIC currents and produce migraine-related pain behavior in animals. Suppressing ASICs significantly reduced Ca^2+^ influx, inhibited calcium overload, and diminished Ca^2+^ toxicity, which exacerbated IS injuries in animal models [62]. These targets are of interest in researching IS headaches. Collecting samples from IS patients or using models may help us to move forward in this area.

Cortical spreading depression (CSD) is widely accepted as an electrophysiologic substrate of migraine aura and a trigger for headaches [63]. In addition, cerebral ischemia is a well-known experimental trigger for CSD and can be recorded during multimodal neuromonitoring in neurocritical care, which serves as a marker for metabolic failure and excitotoxic injury [64]. A complex scenario presents after focal ischemia, in which a gradient of blood flow and impaired metabolism is initiated from the ischemic core and propagates through the classic penumbra into the normally perfused periphery. This produces a reduction in oxidative substrates, decreased tissue ATP, failure of Na^+^/K^+^-ATPase pumps, and increased intracellular Ca^2+^ in neurons and astrocytes [63]. These shared mechanisms promote the close correlations between migraine and cerebral ischemia; however, associated experimental evidence is still limited. Using laser speckle imaging or laser Doppler flow-related blood flow recordings or by measuring changes in the negative direct coupled shift potential [21], CSD might be recorded during the propagation of IS headaches.

## 6. Conclusions

Although IS headaches are an old topic, persistent headaches attributed to IS is a relatively new concept that has not been well documented. Headaches are usually described subjectively, while cerebral infarctions exhibit substantial heterogeneity. There are a multitude of reasons that make it difficult to explore the pathophysiology of headaches attributed to IS based on current existing reports. By identifying and analyzing the changes related to headaches in brain regions and suggesting favorable imaging techniques for researching IS headaches, this review can provide new insights into mechanical studies of IS headaches from structure to function.

## Data Availability

Data sharing not applicable.

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
