# Peer review of "Current Knowledge about Headaches Attributed to Ischemic Stroke: Changes from Structure to Function"

_brainsci, 2023, doi:10.3390/brainsci13071117_

Round 1

Reviewer 1 Report

This review aimed to illustrate the current knowledge of headache attributed to ischemic stroke. The paper is structured into different sections. After the introductive section, the Authors describe the epidemiology, clinical features, risk factors and impact of headache attributed to ischemic stroke. In the third section, they illustrate the current knowledge about the relation between headache and the specific site of infarction. The following section is about imaging techniques currently used to explore headache attributed to ischemic stroke and its pathophysiology. The Authors conclude that headache attributed to stroke is currently little known and modern imaging techniques should be applied to investigate this entity.

The following point should be revised:

Methods section: a methods section is actually completely missing. The Authors should add a section in order to illustrate the review methodology: the data sources they used, the keywords they used; selection criteria of the studies.

 Writing and syntactic mistakes should be corrected as follows:

-line 101: “a research” instead of “an research”

-line 104: “center:” instead of “center,”

-line 115-118: “headache in young patients with mild symptoms might be related to collateral circulation reconstruction, indicating a good prognosis, whilst severe headache indicates tissue compression and structural changes, which indicates poor prognosis” instead of  young patients with mild symptoms prefer to have headache related to collateral circulation reconstruction, indicating a good prognosis; however, patients with severe headache often indicate tissue compression and structural changes, which indicates poor prognosis”

-line 125: “regions” instead of “regions;”

-line 217: the sentence “Neuronal projections from the basal ganglia and hypothalamus to posterior and lateral posterior thalamic nuclei and then to the forebrain“ does not have a verb.

Writing and syntactic mistakes should be corrected as follows:

-line 101: “a research” instead of “an research”

-line 104: “center:” instead of “center,”

-line 115-118: “headache in young patients with mild symptoms might be related to collateral circulation reconstruction, indicating a good prognosis, whilst severe headache indicates tissue compression and structural changes, which indicates poor prognosis” instead of  “young patients with mild symptoms prefer to have headache related to collateral circulation reconstruction, indicating a good prognosis; however, patients with severe headache often indicate tissue compression and structural changes, which indicates poor prognosis”

-line 125: “regions” instead of “regions;”

-line 217: the sentence “Neuronal projections from the basal ganglia and hypothalamus to posterior and lateral posterior thalamic nuclei and then to the forebrain“ does not have a verb.

Author Response

Comments: Methods section: a methods section is actually completely missing. The Authors should add a section in order to illustrate the review methodology: the data sources they used, the keywords they used; selection criteria of the studies. Response: Thank you for your suggestion. This has been added in page 2, line 51-55. Comments:  Writing and syntactic mistakes should be corrected as follows: -line 101: “a research” instead of “an research” -line 104: “center:” instead of “center,” -line 115-118: “headache in young patients with mild symptoms might be related to collateral circulation reconstruction, indicating a good prognosis, whilst severe headache indicates tissue compression and structural changes, which indicates poor prognosis” instead of “young patients with mild symptoms prefer to have headache related to collateral circulation reconstruction, indicating a good prognosis; however, patients with severe headache often indicate tissue compression and structural changes, which indicates poor prognosis” -line 125: “regions” instead of “regions;” -line 217: the sentence “Neuronal projections from the basal ganglia and hypothalamus to posterior and lateral posterior thalamic nuclei and then to the forebrain“ does not have a verb. Response: All your corrections has been accepted in our revised manuscript. The AJE editors helped us to do the proof reading. We believe the writing has been improved and help it is enough for publication.

Reviewer 2 Report

Very interesting and well structured work. As the only suggestion I would try to extend the research in the field of therapy for these conditions: not so much the reperfusion therapy of the stroke as the therapy for the headache that follows or accompanies the event (there are data in the literature, for example, in relation to treatment with steroid acute for extensive infarcts of the structures of the posterior cranial fossa). This is also a chapter that can be developed as a stand-alone article, so I'll understand if you don't include it in this paper.

Author Response

Comments: Very interesting and well structured work. As the only suggestion I would try to extend the research in the field of therapy for these conditions: not so much the reperfusion therapy of the stroke as the therapy for the headache that follows or accompanies the event (there are data in the literature, for example, in relation to treatment with steroid acute for extensive infarcts of the structures of the posterior cranial fossa). This is also a chapter that can be developed as a stand-alone article, so I'll understand if you don't include it in this paper. Response: Thank you for your evaluation. I agree that the therapy for IS headache is a also good topic. We here want to review the happen and mechanisms of IS headache. Therefore, the therapy could be developed separately later.

Reviewer 3 Report

The authors conducted the very nice review on the association between ischemic stroke and headache. The suggestion of mechanical studies of IS headache from structure to function is interesting.

 The paper provided the comprehensive review on epidemiology, clinical characteristics, risk factors, and influence on ischemic stroke headache.

The authors conducted the literature reviews and summarized the findings on epidemiology, clinical characteristics, risk factors, and influence on ischemic stroke headache. 

The review was helpful to understand the association between headache and ischemic stroke and provided the some insights on the structural imaging and functional imaging on exploring the possible ischemic lesions triggered by headache.

Conclusions are consistent with the evidence and arguments presented.

Author Response

Comments: The authors conducted the very nice review on the association between ischemic stroke and headache. The suggestion of mechanical studies of IS headache from structure to function is interesting.  The paper provided the comprehensive review on epidemiology, clinical characteristics, risk factors, and influence on ischemic stroke headache. The authors conducted the literature reviews and summarized the findings on epidemiology, clinical characteristics, risk factors, and influence on ischemic stroke headache.  The review was helpful to understand the association between headache and ischemic stroke and provided the some insights on the structural imaging and functional imaging on exploring the possible ischemic lesions triggered by headache. Conclusions are consistent with the evidence and arguments presented. Response: Thank you for your evaluation.

Round 2

Reviewer 1 Report

Although the paper has been improved the statement added by authors to decribe the methodology used in the review is sitll unsatisfactory. Instead of adding a few sentences in the Introduction a separare paragraph of methods should be included. A description of the selection criteria of the studies is still missing. Moreover, if other sources beyond PubMed have been used should be specified.

Author Response

Comments: Although the paper has been improved the statement added by authors to decribe the methodology used in the review is sitll unsatisfactory. Instead of adding a few sentences in the Introduction a separare paragraph of methods should be included. A description of the selection criteria of the studies is still missing. Moreover, if other sources beyond PubMed have been used should be specified. Response: Thank you for your suggestion. We agree it's important to show how we search and organize the reports here. Unlike Meta-analysis, this has not been performed under certain inclusion and exclusion. Therefore, we described our steps of literature searching during the review. We hope it is fair enough for the publication. Please check it on page 2, lines 54-68.